# RhoGDI2-Mediated Rac1 Recruitment to Filamin A Enhances Rac1 Activity and Promotes Invasive Abilities of Gastric Cancer Cells

**DOI:** 10.3390/cancers14010255

**Published:** 2022-01-05

**Authors:** Hyo-Jin Kim, Ki-Jun Ryu, Minju Kim, Taeyoung Kim, Seon-Hee Kim, Hyeontak Han, Hyemin Kim, Keun-Seok Hong, Chae Yeong Song, Yeonga Choi, Cheol Hwangbo, Kwang Dong Kim, Jiyun Yoo

**Affiliations:** 1Division of Applied Life Science, Gyeongsang National University, Jinju 52828, Korea; jin4477@hanmail.net (H.-J.K.); ryu8650@naver.com (K.-J.R.); kimminju0091@naver.com (M.K.); ta0213@naver.com (T.K.); chenaliii@naver.com (S.-H.K.); entreluzyluz@naver.com (H.H.); gpals8564@naver.com (H.K.); hongs06@naver.com (K.-S.H.); codud929@naver.com (C.Y.S.); younga0702@naver.com (Y.C.); chwangbo@gnu.ac.kr (C.H.); kdkim88@gnu.ac.kr (K.D.K.); 2Research Institute of Life Sciences, Gyeongsang National University, Jinju 52828, Korea; 3Division of Life Science, Gyeongsang National University, Jinju 52828, Korea

**Keywords:** RhoGDI2, Filamin A, Rac1, Trio, gastric cancer, invasion

## Abstract

**Simple Summary:**

Rho GDP dissociation inhibitor 2 (RhoGDI2), a regulator of Rho family GTPase, has been known to promote tumor growth and malignant progression by activating Rac1 in gastric cancer. However, the precise molecular mechanism by which RhoGDI2 activates Rac1 in gastric cancer cells remains unclear. In this study, we found that interaction between RhoGDI2 and Rac1 is a prerequisite for the recruitment of Rac1 to Filamin A. Moreover, we found that Filamin A acts as a scaffold protein that mediates Rac1 activation. Furthermore, we found that Trio, a Rac1-specific GEF, is critical for Rac1 activation in gastric cancer cells. Conclusively, RhoGDI2 increases Rac1 activity by recruiting Rac1 to Filamin A and enhancing the interaction between Rac1 and Trio, which is critical for invasive ability of gastric cancer cells. Our findings suggest that RhoGDI2 might be a potential therapeutic target for reducing gastric cancer cell metastasis.

**Abstract:**

Rho GDP dissociation inhibitor 2 (RhoGDI2), a regulator of Rho family GTPase, has been known to promote tumor growth and malignant progression in gastric cancer. We previously showed that RhoGDI2 positively regulates Rac1 activity and Rac1 activation is critical for RhoGDI2-induced gastric cancer cell invasion. In this study, to identify the precise molecular mechanism by which RhoGDI2 activates Rac1 activity, we performed two-hybrid screenings using yeast and found that RhoGDI2 plays an important role in the interaction between Rac1, Filamin A and Rac1 activation in gastric cancer cells. Moreover, we found that Filamin A is required for Rac1 activation and the invasive ability of gastric cancer cells. Depletion of Filamin A expression markedly reduced Rac1 activity in RhoGDI2-expressing gastric cancer cells. The migration and invasion ability of RhoGDI2-expressing gastric cancer cells also substantially decreased when Filamin A expression was depleted. Furthermore, we found that Trio, a Rac1-specific guanine nucleotide exchange factor (GEF), is critical for Rac1 activation and the invasive ability of gastric cancer cells. Therefore, we conclude that RhoGDI2 increases Rac1 activity by recruiting Rac1 to Filamin A and enhancing the interaction between Rac1 and Trio, which is critical for the invasive ability of gastric cancer cells.

## 1. Introduction

Gastric cancer is one of the most prevalent malignancies worldwide and the third greatest cause of cancer-related deaths, despite the fact that its incidence and mortality have steadily fallen over the last 10 years [1]. There are various therapeutic strategies, including surgery, chemotherapy, and radiotherapy, amongst others available for gastric cancer patients. However, the prognosis remains unsatisfactory, with high mortality [2]. Despite advances in understanding and improved treatments over the last decade, metastasis remains the leading cause of death in cancer patients. Determining the molecular pathways driving gastric cancer metastatic spread could lead to novel treatment targets.

Most Rho GTPases work as molecular switches, cycling between a cytosolic inactive GDP-bound state and a membrane-associated active GTP-bound state, similar to other members of the Ras superfamily of GTPases. Guanine nucleotide exchange factors (GEFs) can activate Rho GTPases by enhancing the release of bound GDP and the subsequent binding of ambient GTP [3]. Inactivation occurs when GTP hydrolysis is catalyzed by GTPase-activating proteins (GAPs), which convert Rho GTPases to the inactive GDP-bound form [4]. Rho GTPases are also regulated at a different level by another type of protein: Rho GDP dissociation inhibitors (RhoGDIs). Only three human RhoGDIs have been found, compared to the large number of GEF and GAP regulators (each with about 60 members): RhoGDI1 (also known as RhoGDI or RhoGDIα), RhoGDI2 (Ly-GDI, D4-GDI, or RhoGDIβ), and RhoGDI3 (RhoGDIγ) [5,6,7]. RhoGDIs were originally considered to be negative regulators of Rho GTPases because they bind to the majority of Rho GTPases in the cytoplasm and keep them inactive, preventing any interaction with the target effector proteins [8,9]. However, RhoGDIs have been associated with the active forms of Rho, Rac, and Cdc42 in other reports, suggesting that they also act as positive regulators of Rho GTPases [10,11].

Accumulating evidence has demonstrated that RhoGDI2 is expressed differently in different types of human cancer at the mRNA and/or protein levels [12,13,14,15], and it may contribute to aggressive phenotypes by deregulating the Rho GTPase signaling pathway, making it a promising target for cancer therapy [16]. RhoGDI2 expression has previously been shown to be associated with tumor progression and metastatic potential in gastric cancer [17]. Furthermore, we found that RhoGDI2 regulates Rac1 activity and that Rac1 activation is required for RhoGDI2-induced gastric cancer cell invasion [18].

In this study, we aimed to uncover the molecular mechanism by which RhoGDI2 activates Rac1 activity and promotes metastatic characteristics of gastric cancer cells. Our findings might help unravel the role of RhoGDI2 in promoting gastric cancer metastasis, thereby providing potential targets for treatment of gastric cancer.

## 2. Materials and Methods

### 2.1. Cell Culture

All of the cells utilized in this study were obtained from the Korean Cell Line Bank (KCLB). The HEK293T cells were grown in DMEM (Invitrogen, Waltham, MA, USA) supplemented with 10% FBS and 1% penicillin and streptomycin. The SNU484 and MKN28 human gastric cancer cell lines were grown in RPMI (Invitrogen, Waltham, MA, USA) supplemented with 10% FBS and 1% penicillin and streptomycin. In our previous report, we described the SNU484 cells that were stably transfected with RhoGDI2 [17].

### 2.2. Gene Cloning and Transfection

The full-length human RhoGDI2, Filamin A and Rac1 genes were sub-cloned into a pDONR207 vector (Entry vector) using the Gateway Cloning System (Invitrogen, Waltham, MA, USA) following the manufacturer’s instructions. The entry clones were converted into several destination vectors, pDEST-GFP-C, pDEST-FLAG-C and pDEST-HA-C. For transient transfection, cells were seeded in 6-well or 100-mm-diameter dishes for 24 h and transfected with the indicated plasmid by using Fugene 6 transfection reagent (Roche, Basel, Switzerland) following the manufacturer’s instructions. After 48 h, the cells were harvested and used for Western blot analysis.

### 2.3. Antibodies and Western Blot Analysis

For Western blot analysis, cells were harvested after defined time and lysed in lysis buffer (20 mM Tris (pH 7.4), 2 mM EDTA, 150 mM sodium chloride, 1 mM sodium deoxycholate, 1% Triton X-100, 10% glycerol, 2 pills protease inhibitor cocktail (Roche, Basel, Switzerland) on ice for 1 h and centrifuged at 13,000 rpm for 15 min. Cell lysates were subjected to 7.5–15% SDS-PAGE and transferred to polyvinylidene fluoride (PVDF) membrane (Amersham Bioscience, Buckinghamshire, UK). The membrane was incubated overnight with specified primary antibodies at 4 °C. Membranes were incubated with matching HRP-conjugated secondary antibodies for 1 h at room temperature after washing with TBST (TBS containing 0.1% Tween-20). Blots were developed with an enhanced chemiluminescence (ECL, Amersham Bioscience, Buckinghamshire, UK) reaction as directed by manufacturer’s instructions. The antibodies used in this study were as follows: rabbit anti-RhoGDI2 antibody (purchased from Spring Bioscience, Pleasanton, CA, USA, E2434), mouse anti-Filamin A (Millipore, Burlington, MA, USA, MAB1680), mouse anti-Rac1 antibody (BD Transduction Laboratories, San Jose, CA, USA, 610651), rabbit anti-Trio antibody (Bethyl Laboratories, Montgomery, TX, USA, A304-269A), mouse anti-GFP antibody, mouse anti-HA antibody (Santa Cruz, Dallas, TX, USA, sc-9996 and sc-7392, respectively), and mouse anti-α-tubulin antibody (Sigma, St. Louis, MO, USA, T6199).

### 2.4. Yeast Two-Hybrid Screening

The RhoGDI2(WT) and RhoGDI2 Δ20 genes were cloned into pGBK7, which encodes a GAL4 DNA binding domain (BD), while the Rac1 and FLNA/Rod2 genes were cloned into pGADT7, which encodes an activation domain (AD). The corresponding constructs were co-transformed into the yeast strain AH109 to evaluate the interactions between these proteins. For growth selection, the transformants were grown in synthetic dropout (SD) medium lacking Leu, Trp, and His (Leu-/Trp-/His-).

### 2.5. Immunoprecipitation

To confirm the interaction between the two proteins, immunoprecipitation was performed. The cells were lysed in lysis buffer, vortexed, and kept on ice for 30 min. Protein A/G beads (Santa Cruz, Dallas, TX, USA) were used to pre-clear the lysates, which were subsequently incubated with the indicated antibodies for 2 h at 4 °C before being incubated with beads overnight at 4 °C with moderate mixing. After that, the beads were eluted with 30 μL of 2 × SDS sample buffer before being analyzed using a Western blot.

### 2.6. RNA Interference Experiments and Transfection

The RhoGDI2-overexpressing SNU-484 and MKN28 cells were stably transfected with shRNA-expressing lentiviral vector targeting Filamin A. Transfection was performed by adding shRNA particles along with 5 μg/mL of hexadimethrine bromide (Polybrene, Santa Cruz, Dallas, TX, USA) into cell suspensions in 6-well plates. Clones stably expressing the shRNA plasmids were selected and placed in 500 ng/mL puromycin for 5 days, and the cells were stocked and confirmed for protein expression using Western blot analysis. The shRNA target sequences used in this study were as follows; shFLAN-1; 5′-CCGGCCCGCCTGTCACTGCAGCTGCCTCGAGGCAGCTGCAGTGACAGGCGGGTTTTTG-3′, shFLNA-2; 5′-CCGGCCCACCCACTTCACAGTAAATCTCGAGTTTACTGTGAAGTGGGTGGGTTTTTG-3′. Two different siRNA oligo duplexes for targeting the Trio gene were purchased from Bioneer. The sequence was as follows: siTrio-1; 5′-CCUGAACAUCAAGAGUGUUCA-3′, shTrio-2; 5′-AUGAUGAAUCAAGAGUUCAUC-3′. Transient transfection of each siRNA oligo duplex was accomplished using LipofectAMINE Plus Reagent (Invitrogen, Waltham, MA, USA) following the instructions of the manufacturers. After incubation for 48 h, the cells were harvested and the efficiency of each siRNA oligo duplex was confirmed by Western blotting using anti-Trio antibody.

### 2.7. Rac Activity Assay

Rac GTPase activity was determined using Rac/Cdc42 Assay Reagent (Upstate Biotechnology, Lake Placid, NY, USA) according to the manufacturer’s instructions. 1 mL MLB buffer was used to lyse the cells. PAK-1-agarose was added to an equal amount of cell lysates right away and incubated at 4 °C for 1 h. After centrifuging the agarose beads, the bead pellet was washed in MLB buffer for three times. Finally, the bead pellet was suspended in Laemmli sample buffer and Western blot analysis was performed using anti-Rac1 antibody.

### 2.8. Migration Assay

4.0 × 10^4^ cells in 70 μL of medium with 10% FBS were cultured in Culture-Insert (Ibidi, Munich, Germany) for wound healing assays. To reduce the effect of cell growth, the cells were pretreated with 10 µg/mL mitomycin C (Sigma, St. Louis, MO, USA) for 2 h before being washed with culture media. Cells were cultured with fresh medium after the Culture-Insert was removed, and images of the migration assay were taken at 0 and 24 h using a phase-contrast microscope with a digital camera.

### 2.9. Invasion Assay

Invasion assays were carried out using QCM™ 24-Well Cell Invasion Assay kit (Chemicon, Temecula, CA, USA) according to the manufacturer’s instructions. 2.5 × 10^5^ cells were suspended in 250 μL of serum free media and cultured for 24 h in the insert (8 μm pore size). In the lower chamber, 500 μL of suitable medium containing 20% FBS was inserted. Cells/medium that remained on the top side of the insert after incubation were pipetted away. The invasion chamber insert was moved to a clean well containing pre-warmed Cell Detachment Solution, and incubated at 37 °C for 30 min. The insert was taken out of the well. Each well was incubated for 15 min at room temperature with 75 μL of Lysis Buffer/Dye solution (CyQuant GR Dye 1:75 with 4 × Lysis Buffer). 200 μL of the mixture were poured into a 96-well plate and analyzed using a fluorescence plate reader with a 480/520 nm filter set.

### 2.10. Proliferation Assay

Cells were cultured at a concentration of 1 × 10^4^ cells per well in a 6-well plate. Trypsinized cells were resuspended in 3 mL of suitable media after incubation for 1 to 4 days. For 5 min, cell suspensions were centrifuged at 1000 rpm. The pellets were resuspended in 1 mL of the appropriate medium. After trypan blue staining, a hemocytometer was used to count the viable cells.

### 2.11. Statical Analysis

Statistical analysis was performed using the PASW Statistics 18.0 software (IBM Corporation, Somers, NY, USA). Data represent the means ± SD. The significance of the differences was determined using the chi-square test. A Student’s *t*-test was applied for pairwise comparisons. For multiple comparisons, data were tested by one-way ANOVA, and subsequently using the Dunnett post hoc test. *p*-values < 0.05 were considered to be statistically significant.

## 3. Results

### 3.1. Interaction between RhoGDI2 and Rac1 Is a Prerequisite for RhoGDI2 to Interact with Filamin A

To investigate the molecular mechanism by which RhoGDI2 activates Rac1, we searched for novel proteins that interact with RhoGDI2 via yeast two-hybrid screening; however, we only identified Rac1, which is already known to interact with RhoGDI2 [18] (Figure 1A). Thus, we hypothesized that the domain of RhoGDI2 interaction with Rac1 could prevent RhoGDI2 from binding to other proteins. Therefore, we performed yeast two-hybrid screening again using a construct where the N-terminal 20 amino acids of RhoGDI2 that bind with Rac1 [19,20] were removed (RhoGDI2Δ20). Furthermore, we discovered the Rod-2 domain of Filamin A (FLNA/Rod-2, Figure 1B) as a new binding partner for RhoGDI2Δ20 in yeast (Figure 1A). Next, we verified these interactions in cells and found that the binding between Rac1 and wild-type (WT) RhoGDI2 was stronger than that between Rac1 and RhoGDI2Δ20 (Figure 1C), whereas the binding between FLNA/Rod-2 and RhoGDI2Δ20 was stronger than that between FLNA/Rod-2 and WT-RhoGDI2 (Figure 1D) in HEK293T cells.

Based on the above results, we hypothesized that if Rac1 binds to RhoGDI2, a conformational change in RhoGDI2 is induced, and the domain of RhoGDI2 that binds with FLNA can be exposed (Figure 1E). To verify this hypothesis, RhoGDI2 with or without Rac1 was transiently transfected into HEK293T cells, and a significant increase in the interaction between RhoGDI2 and endogenous FLNA was found when Rac1 was co-expressed (Figure 1F). These results suggest that the interaction between RhoGDI2 and Rac1 is a prerequisite for RhoGDI2 to interact with FLNA.

### 3.2. RhoGDI2 Plays an Important Role in the Interaction between Rac1 and Filamin A and Rac1 Activation in Gastric Cancer Cells

Since FLNA is known to interact with GEFs capable of activating Rac1 [21,22], we presumed RhoGDI2 would bind to Rac1 and then transfer to FLNA to activate Rac1. To test this hypothesis, we first investigated whether the expression of RhoGDI2 increases the interaction between Rac1 and FLNA and found that the interaction of Rac1 with endogenous FLNA was markedly increased in HEK293T cells when RhoGDI2 was co-expressed (Figure 2A). As shown in Figure 1, when compared to Rac1 and WT-RhoGDI2, the binding between Rac1 and RhoGDI2Δ20 was markedly reduced; therefore, we assumed that RhoGDI2Δ20 could not increase the interaction between Rac1 and FLNA even if it was co-expressed (Figure 2B). Interestingly, RhoGDI2 could enhance the binding of Rac1 to endogenous FLNA, but RhoGDI2Δ20 could not (Figure 2C). These results suggest that RhoGDI2 plays an important role in transferring Rac1 to FLNA after binding to Rac1.

Next, we investigated whether RhoGDI2 could enhance the interaction between Rac1 and FLNA in gastric cancer cells. For this experiment, we used SNU484 cells, a primary gastric cancer cell line that does not express RhoGDI2 and has low invasive ability, and MKN28 cells, which express large amounts of RhoGDI2 and have high metastatic capacity [17]. Overexpression of RhoGDI2 in SNU484 cells caused a significant increase to invasiveness in vitro and to tumor metastasis in vivo [17]. In our previous report, we have already shown that Rac1 activity was significantly increased in RhoGDI-overexpressing SNU484 cells and decreased in RhoGDI2-depleted MKN28 cells compared with its expression in control cells [18]. We also confirmed the interaction of RhoGDI2 with Rac1 in RhoGDI2-overexpressing SNU484 and MKN28 cells (Appendix A). We found that the interaction of endogenous Filamin A with Rac1 was only seen in MKN28 cells, but not in SNU484 cells (Figure 2D). We demonstrated that the interaction between endogenously expressed Rac1 and FLNA was markedly increased in WT RhoGDI2-overexpressing SNU484 cells, but not in RhoGDI2Δ20-overexpressing cells (Figure 2E). In MKN28 cells, the decrease in RhoGDI2 expression significantly reduced the interaction between FLNA and Rac1 (Figure 2F). We also saw that Rac1 activity was significantly increased in WT RhoGDI2-overexpressing SNU484 cells, but not in RhoGDI2Δ20-overexpressing cells (Figure 2G). These results are consistent with our previous results showing that the Rac1 activity was significantly reduced in RhoGDI2-depleted MKN28 cells [18]. All of our results clearly suggest that RhoGDI2 plays an important role in the interaction between Rac1 and Filamin A and Rac1 activation in gastric cancer cells.

### 3.3. Filamin A Is Required for Rac1 Activation and the Invasive Ability of Gastric Cancer Cells

To determine whether FLNA is required for Rac1 activation in gastric cancer cells, FLNA expression was depleted in MKN28 and WT RhoGDI2-overexpressing SNU484 cells (SNU484/GDI2), where Rac1 was already activated with high expression of RhoGDI2. We found that decreasing FLNA expression markedly reduced Rac1 activity in SNU484/GDI2 and MKN28 cells (Figure 3A). Next, to investigate the migratory properties of FLNA-depleted cells, we performed a wound healing assay. After 24 h, it was found that FLNA-depleted SNU484/GDI2 and MKN28 cells had moved into the scratch wound less than the control cells (Figure 3B). In an in vitro invasion study, we found that the number of invasive FLNA-depleted SNU484/GDI2 and MKN28 cells was significantly lower than in the control cells (Figure 3C). However, under the same growth conditions, all cells grew at a similar pace (Figure 3D), demonstrating that the reduced migration and invasion shown in RhoGDI2-expressing gastric cancer cells in response to FLNA depletion was independent of their growth rates.

### 3.4. Trio Is Critical for Rac1 Activation and Invasive Ability of Gastric Cancer Cells

There are several GEFs that can activate Rac1. Among them, Trio, a Rac1-specific GEF, has been known to interact with FLNA [21]. We found that Trio was bound to FLNA in SNU484 and MKN28 gastric cancer cell lines (Figure 4A). We next investigated whether RhoGDI2 could enhance the interaction between Rac1 and Trio in gastric cancer cells. We observed that the interaction between endogenously expressed Rac1 and Trio was markedly increased in WT RhoGDI2-overexpressing SNU484 cells, but not in RhoGDI2Δ20-overexpressing cells (Figure 4B). In MKN28 cells, the decrease in RhoGDI2 expression significantly reduced the interaction between Rac1 and Trio (Figure 4C). These results suggest that RhoGDI2 expression is essential for the interaction between Trio and Rac1 in gastric cancer cells.

To determine whether Trio is critical for Rac1 activation in gastric cancer cells, we depleted Trio expression in SNU484/GDI2 and MKN28 cells, which led to marked reduction of Rac1 activity in these cells (Figure 4D). Next, we investigated whether Trio depletion could alter the migratory properties of these cells and found that a decreased number of Trio-depleted SNU484/GDI2 and MKN28 cells migrated into the scratch wound than the control cells (Figure 4E). Similarly, in an in vitro invasion assay, we detected a substantial decrease in the number of invasive Trio-depleted SNU484/GDI2 and MKN28 cells relative to control cells (Figure 4F). However, all cells exhibited similar growth rates under the same growth conditions (Figure 4G). All of these results suggest that Trio is critical for Rac1 activation and invasive ability of gastric cancer cells.

### 3.5. Expression of RhoGDI2 and Filamin A could Be a Poor Prognostic Marker for Gastric Cancer Patients

To address the clinical significance of RhoGDI2 and FLNA expression in gastric cancer patients, we analyzed public microarray data sets (GSE54129, GSE26901 and GSE26942), and found that expression of RhoGDI2 and FLNA was increased in gastric cancer tissues compared to normal tissues (Figure 5A,B), and increased in advanced-stage gastric cancer tissues (tumor stages III and IV) more than in early-stage gastric cancer tissues (tumor stages I and II) (Figure 5C,D). We also found that higher expression of both RhoGDI2 and FLNA was associated with poor prognosis in gastric cancer patients (Figure 5E). These results suggest that expression of RhoGDI2 and FLNA could be a poor prognostic marker in gastric cancer patients.

## 4. Discussion

RhoGDI2 is known to be differentially expressed in various types of human cancer [16]. For instance, when compared to non-Hodgkin’s lymphoma cells, RhoGDI2 expression is downregulated in Hodgkin’s lymphoma cells, and a reduction of RhoGDI2 expression can contribute to resistance to cell death in Hodgkin’s lymphoma cells [23]. Similarly, a bladder cancer cell line T24 expresses more RhoGDI2 than T24T, a more aggressive lineage, where its forced expression attenuated metastatic ability of cancer cells [12,24]. These findings suggest that RhoGDI2 is a metastatic inhibitor in bladder cancer cells. On the other hand, RhoGDI2 has been linked to advanced malignancies and increased metastatic abilities in ovarian and breast cancer cells [14,15]. In our previous study, we demonstrated that elevated expression of RhoGDI2 is linked to advanced gastric cancer and the metastatic abilities of gastric cancer cells [17].

The explanation of the disparity in RhoGDI2′s participation in different cancers remains unclear; however, the dual role of RhoGDI in controlling the activity of Rho GTPase during cancer progression may be one of the plausible reasons. RhoGDI was identified as a negative regulator of Rho GTPases at first [8,9]. RhoGDI caused the loss of Rho-dependent cell activity, such as cytoskeletal activity and motility, when exogenously introduced into cells [25]. However, recent reports suggest that RhoGDI1 has a favorable role in Rho GTPase function: it may function as an escort protein, directing Rho GTPases to the membrane, which is required for GTPases to associate with downstream effector proteins. For example, Rac1 must form a complex with RhoGDI to regulate NADPH oxidase activity in neutrophils [26]. Similarly, RhoGDI1, despite being a negative regulator of Cdc42 activation, plays an essential role in Cdc42-mediated cellular transformation in NIH-3T3 fibroblast cells by delivering Cdc42 to a specific cellular location [27]. Following these studies, we investigated the activation status of Rho GTPases in RhoGDI2-expressing and -depleted cells, and discovered that RhoGDI2 activates Rac1 exclusively in gastric cancer cells. It is presumable that RhoGDI2 may act as a Rac1 positive regulator in gastric cancer cells [18]. Rac and Rho family proteins play a key role in cell motility regulation, specifically through the formation of lamellapodia [28]. Constitutively active Rac1 V12 overexpression-induced tumors in mice were phenotypically similar to human Kaposi’s sarcoma [29]. Rac1 activation has been found to have an important function in facilitating type of assisted cell movement [30]. These reports suggest that Rac1 is a protein that increases cancer metastasis. Keeping in mind that enhancing Rac1 activity by RhoGDI2 may be important for the metastasis of gastric cancer, we have identified its mechanism.

In this study, we performed a yeast two-hybrid screening experiment to find a new binding protein to characterize the role of RhoGDI2 in gastric cancer. Filamin A (FLNA), an actin binding protein, is a 280 kD protein that cross-links actin filaments and participates in membrane protein immobilization for the actin cytoskeleton [31,32,33,34,35]. It is known to act as a molecular scaffold for intracellular signals [36,37]. FLNA interacts with small GTPases, including Rac, Rho, Cdc42 and RalA, and also with GTPase-related proteins [21,38,39]. Therefore, we theorized that FLNA could be a scaffold protein that mediates Rac1 activation via RhoGDI2 in gastric cancer cells. Indeed, when RhoGDI2 mutated or its expression was depleted, binding between Rac1 and FLNA decreased and the activity of Rac1 reduced. Additionally, Rac1 activity was reduced in FLNA-depleted gastric cancer cells. Through these findings, we concluded that FLNA is critical for gastric cancer metastasis by enhancing Rac1 activity. Consistent with our results, Song et al. identified the interaction between the Rod-2 domain of Filamin A and the 120–146 amino acid residues of RhoGDI2 [40]. Based on these findings, we could predict that the N-terminal domain of RhoGDI2 interacts with 120–146 amino acid residues intramolecularly, and when Rac1 binds to the N-terminal domain of RhoGDI2, a conformational change occurs, exposing 120–146 amino acid residues and recruiting Rac1 to Filamin A to activate Rac1.

The GTPases, including Rac1, are active when bound to GTP and inactive when bound to GDP. GEFs are essential for the activation of small GTPases because GDP dissociates from inactive GTPases very slowly. Trio, one of the Rac1-specific GEFs, has been known to interact with FLNA [21]. In this study, we found that RhoGDI2 could enhance the interaction between Rac1 and Trio in gastric cancer cells, which is critical for Rac1 activation and invasive ability in gastric cancer cells (Figure 6).

## 5. Conclusions

In conclusion, our study reveals a novel mechanism for RhoGDI2-induced Rac1 activation in gastric cancer cells. We have identified the precise molecular mechanism by which RhoGDI2 recruits Rac1 to FLNA and increases Rac1 activity, which is critical for gastric cancer metastasis.

## Figures and Tables

**Figure 1 cancers-14-00255-f001:**
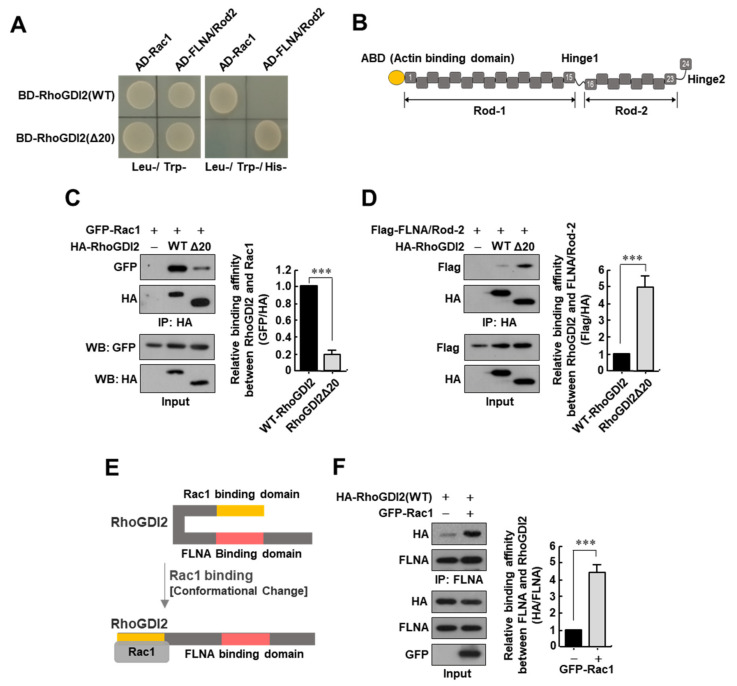
Interaction between RhoGDI2 and Rac1 is a prerequisite for RhoGDI2 to interact with Filamin A. (**A**) Yeast two-hybrid experiments to see the interactions between wild type RhoGDI2 or RhoGDI2Δ20 and the Rac1 or Rod2 domain of Filamin A. (**B**) Schematic diagram of Filamin A structure. ABD; actin-binding domain, Rod-1 and Rod-2; immunoglobulin-like tandem repeat domains. (**C**) Interaction between exogenous Rac1 with wild type RhoGDI2 or RhoGDI2Δ20. HEK293T cells transfected with indicated constructs were immunoprecipitated and analyzed by Western blot (**left**). The data are representative of three independent experiments, and relative GFP and HA levels were quantified using ImageJ software (Version 1.53n, NIH, Bethesda, MD, USA) (**right**). ***, *p* < 0.001 as determined by *t*-test. The uncropped immunoblot images can be found in Appendix A. (**D**) Interaction between exogenous Rod2 domain of Filamin A (FLNA/Rod-2) and wild type RhoGDI2 or RhoGDI2Δ20. HEK293T cells transfected with indicated constructs, immunoprecipitated and analyzed by Western blot (**left**). The data are representative of three independent experiments and relative Flag and HA levels were quantified using ImageJ software (**right**). ***, *p* < 0.001 as determined by *t*-test. The uncropped immunoblot images can be found in Appendix A. (**E**) Schematic diagram of the conformational changes in RhoGDI2 after binding to Rac1. (**F**) Interaction between endogenous Filamin A with exogenous RhoGDI2 in the presence or absence of Rac1. HEK293T cells transfected with indicated constructs were immunoprecipitated and analyzed by Western blot (**left**). The data are representative of three independent experiments and relative FLNA and HA levels were quantified using ImageJ software (**right**). ***, *p* < 0.001 as determined by *t*-test. The uncropped immunoblot images can be found in Appendix A.

**Figure 2 cancers-14-00255-f002:**
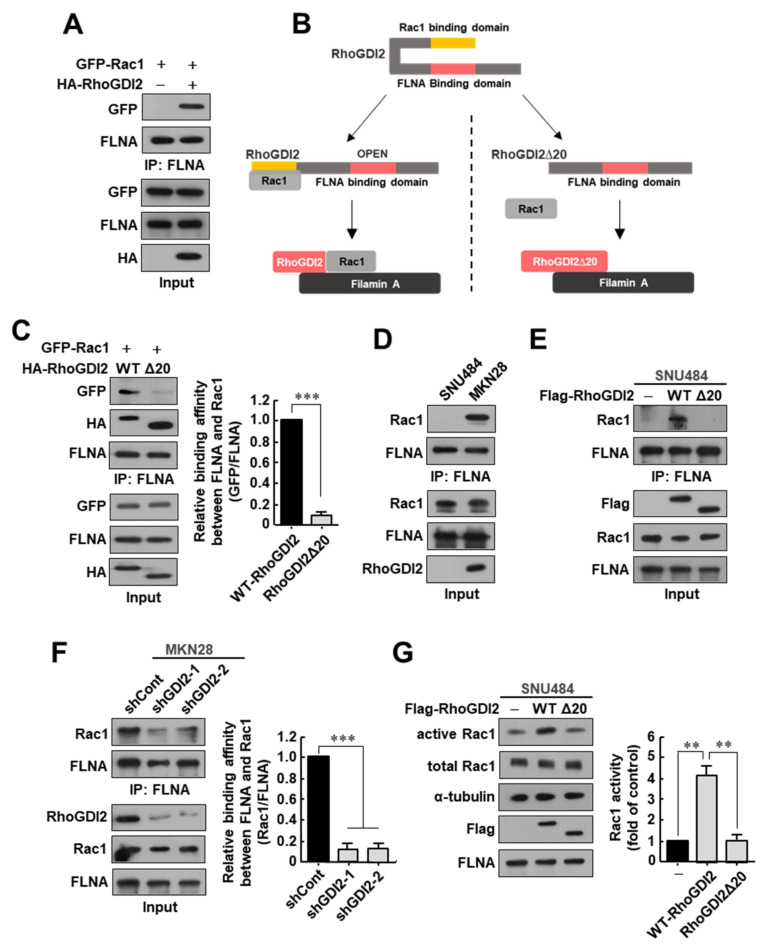
RhoGDI2 plays an important role in the interaction between Rac1 and Filamin A and Rac1 activation in gastric cancer cells. (**A**) Interaction between endogenous Filamin A with exogenous Rac1 in the presence or absence of RhoGDI2. HEK293T cells transfected with indicated constructs were immunoprecipitated and analyzed by Western blot. The uncropped immunoblot images can be found in Appendix A. (**B**) Schematic diagram of the conformational changes in wild type RhoGDI2 or RhoGDI2Δ20 after binding to Rac1. (**C**) Interaction between endogenous Filamin A with exogenous Rac1 in the presence of wild type RhoGDI2 or RhoGDI2Δ20. HEK293T cells transfected with indicated constructs were immunoprecipitated and analyzed by Western blot (**left**). The data are representative of three independent experiments and relative FLNA and GFP levels were quantified using ImageJ software (**right**). ***, *p* < 0.001 as determined by *t*-test. The uncropped immunoblot images can be found in Appendix A. (**D**) Interaction between endogenous Filamin A with endogenous Rac1 in SNU484 and MKN28 gastric cancer cell lines. SNU484 and MKN28 cell lysates were immunoprecipitated and analyzed by Western blot. The uncropped immunoblot images can be found in Appendix A. (**E**) Interaction between endogenous Filamin A with endogenous Rac1 in the presence of wild type RhoGDI2 or RhoGDI2Δ20 in SNU484 gastric cancer cells. SNU484 cells transfected with indicated constructs were immunoprecipitated and analyzed by Western blot. The uncropped immunoblot images can be found in Appendix A. (**F**) Interaction between endogenous Filamin A with endogenous Rac1 in RhoGDI2-depleted MKN28 gastric cancer cells. MKN28 cells transfected with indicated constructs were immunoprecipitated and analyzed by Western blot (**left**). The data are representative of three independent experiments and relative FLNA and Rac1 levels were quantified using ImageJ software (**right**). ***, *p* < 0.001 as determined by *t*-test. The uncropped immunoblot images can be found in Appendix A. (**G**) Rac1 activity of SNU484 cells transfected with wild type RhoGDI2 or RhoGDI2Δ20. SNU484 cells transfected with indicated constructs were precipitated with a PAK1-agarose and western blotted using anti-Rac1 antibody (**left**). The data are representative of three independent experiments and ImageJ software was used to calculate the relative active Rac1 levels (**right**). For normalization, total Rac1 and α-tubulin expressions were used as controls. **, *p* < 0.01 as determined by *t*-test. The uncropped immunoblot images can be found in Appendix A.

**Figure 3 cancers-14-00255-f003:**
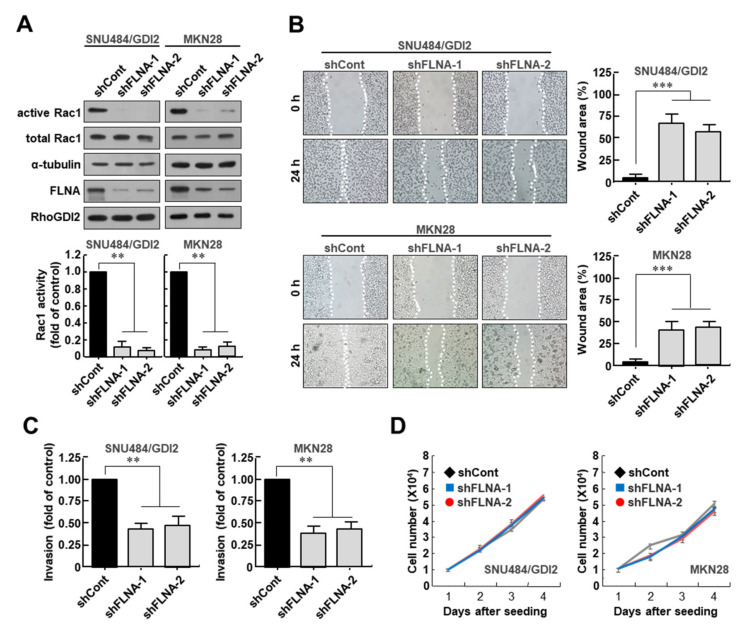
Filamin A is required for Rac1 activation and invasive ability of gastric cancer cells. (**A**) Effect of Filamin A depletion on Rac1 activity in RhoGDI2-overexpressing SNU484 and MKN28 gastric cancer cell lines. Filamin A-depleted cell lysates were precipitated with a PAK1-agarose and analyzed by Western blot using anti-Rac1 antibody (upper). The data are representative of three independent experiments and ImageJ software was used to calculate the relative active Rac1 levels (lower). For normalization, total Rac1 and α-tubulin expressions were used as controls. **, *p* < 0.01 as determined by *t*-test. The uncropped immunoblot images can be found in Appendix A. (**B**) Effect of Filamin A depletion on migration ability of RhoGDI2-overexpressing SNU484 and MKN28 gastric cancer cells. A wound-healing assay was used to examine the migration ability of Filamin A-depleted SNU484/GDI2 and MKN28 cells, with wound closure visualized using phase-contrast microscopy (**left**). WimScratch software (Wimasis) was used to measure the wound areas. The percentage of the wound area is expressed as the means ± SD of three separate experiments (**right**). ***, *p* < 0.001 as determined by *t*-test. (**C**) Effect of Filamin A depletion on invasive ability of RhoGDI2-overexpressing SNU484 and MKN28 gastric cancer cell lines. Filamin A-depleted SNU484/GDI2 and MKN28 cells were cultured onto matrix-coated upper chambers and the number of invading cells was measured after 24 h. The data are presented as the means ± SD of three separate experiments, carried out in triplicate. **, *p* < 0.01 as determined by *t*-test. (**D**) Effect of Filamin A depletion on proliferation rate of RhoGDI2-overexpressing SNU484 and MKN28 gastric cancer cell lines. The indicated cells were cultured at a concentration of 1 × 10^4^ cells per well in a 6-well plate. After trypan blue staining, the viable cells were counted with a hemocytometer after 1 to 4 days of incubation.

**Figure 4 cancers-14-00255-f004:**
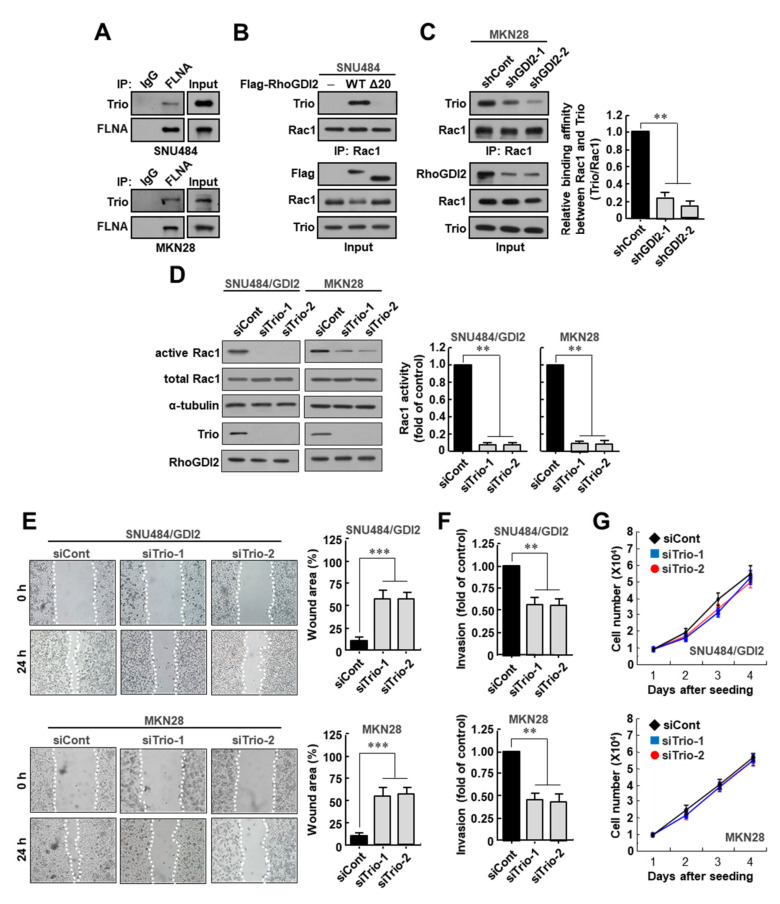
Trio is critical for Rac1 activation and invasive ability of gastric cancer cells. (**A**) Interaction between endogenous Filamin A with endogenous Trio in SNU484 and MKN28 cells. SNU484 and MKN28 cells were immunoprecipitated and analyzed by Western blot. The uncropped immunoblot images can be found in Appendix A. (**B**) Interaction between endogenous Rac1 with endogenous Trio in the presence of wild type RhoGDI2 or RhoGDI2Δ20 in SNU484 gastric cancer cells. SNU484 cells transfected with indicated constructs were immunoprecipitated and analyzed by Western blot. The uncropped immunoblot images can be found in Appendix A. (**C**) Interaction between endogenous Rac1 with endogenous Trio in RhoGDI2-depleted MKN28 gastric cancer cells. MKN28 cells transfected with indicated constructs were immunoprecipitated and analyzed by Western blot (**left**). The data are representative of three independent experiments, and relative Rac1 and Trio levels were quantified using ImageJ software (**right**). **, *p* < 0.01 as determined by *t*-test. The uncropped immunoblot images can be found in Appendix A. (**D**) Effect of Trio depletion on Rac1 activity in RhoGDI2-overexpressing SNU484 and MKN28 gastric cancer cell lines. SNU484/GDI2 and MKN28 cells transfected with indicated constructs were precipitated with a PAK1-agarose and western blotted using anti-Rac1 antibody (**left**). The data are representative of three independent experiments and ImageJ software was used to calculate the relative active Rac1 levels (**right**). For normalization, total Rac1 and α-tubulin expressions were used as controls. **, *p* < 0.01 as determined by *t*-test. The uncropped immunoblot images can be found in Appendix A. (**E**) Effect of Trio depletion on migration ability of RhoGDI2-overexpressing SNU484 and MKN28 gastric cancer cell lines. A wound-healing assay was used to examine the migration ability of Trio-depleted SNU484/GDI2 and MKN28 cells, with wound closure visualized using phase-contrast microscopy (**left**). WimScratch software (Ibidi, Munich, Germany) was used to measure the wound areas. The percentage of the wound area is expressed as the means ± SD of three separate experiments (**right**). ***, *p* < 0.001 as determined by *t*-test. (**F**) Effect of Trio depletion on invasive ability of RhoGDI2-overexpressing SNU484 and MKN28 gastric cancer cell lines. SNU484/GDI2 and MKN28 cells depleted in Trio were cultured onto matrix-coated upper chambers and the number of invading cells was measured after 24 h. The data are presented as the means ± SD of three separate experiments, carried out in triplicate. **, *p* < 0.01 as determined by *t*-test. (**G**) Effect of Trio depletion on proliferation rate of RhoGDI2-overexpressing SNU484 and MKN28 gastric cancer cell lines. The indicated cells were cultured at a concentration of 1 × 10^4^ cells per well in a 6-well plate. After trypan blue staining, the viable cells were counted with a hemocytometer after 1 to 4 days of incubation.

**Figure 5 cancers-14-00255-f005:**
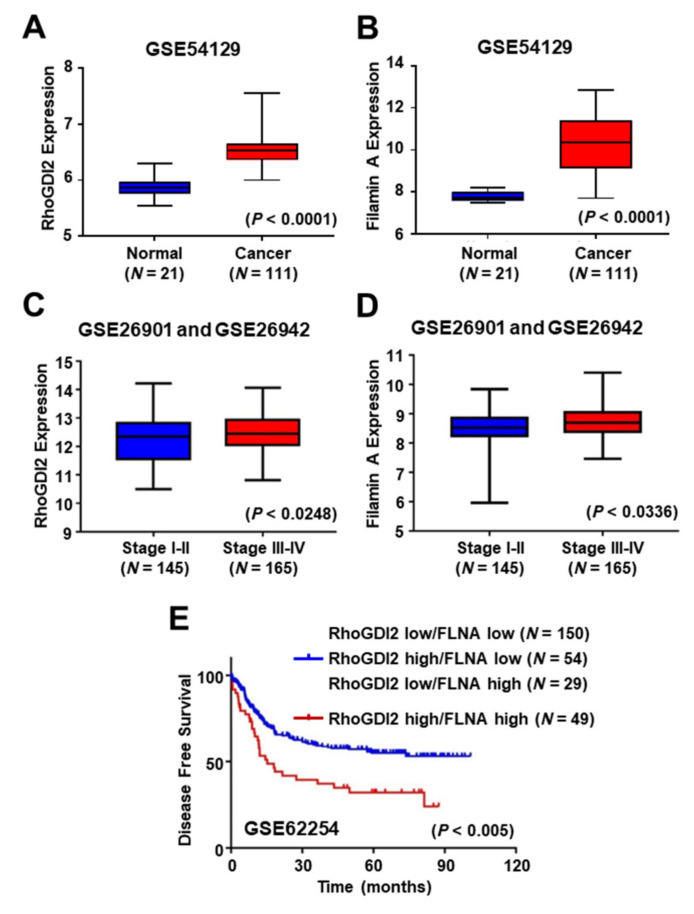
Expression of RhoGDI2 and Filamin A could be poor prognostic markers in gastric cancer patients. (**A**,**B**) Comparison of RhoGDI2 (**A**) and Filamin A (**B**) expression levels between normal and gastric cancer patient tissues from the GEO data set (GSE54129). The interquartile range is shown by the boxes, the median is represented by the center, and the whiskers represent the minimum and maximum values. The unpaired, two-tailed Student’s *t*-test was used to calculate *p*-values. (**C**,**D**) Comparison of RhoGDI2 (**C**) and Filamin A (**D**) expression levels in different stages of gastric cancer patients from GEO data sets (GSE26901 and GSE26942). The interquartile range is shown by the boxes, the median is represented by the center, and the whiskers represent the minimum and maximum values. The unpaired, two-tailed Student’s *t*-test was used to calculate *p*-values. (**E**) Correlation of RhoGDI2 and Filamin A expression levels with disease free survival in 282 gastric cancer patients using a publicly available microarray data set (GSE62254). A Kaplan–Meier plot analysis showed disease free survival depending on the expression level of both RhoGDI2 and Filamin A. A log-rank test was used to calculate *p*-values.

**Figure 6 cancers-14-00255-f006:**
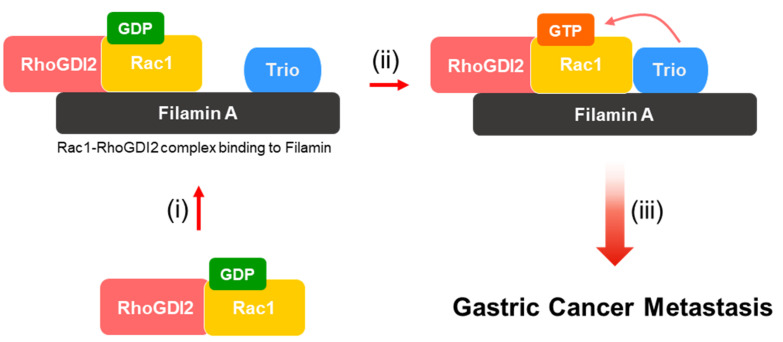
Proposed model to illustrate how activation of Rac1 by RhoGDI2 may lead to gastric cancer metastasis. (i) Interaction between RhoGDI2 and Rac1 is a prerequisite for the recruitment of Rac1 to Filamin A. (ii) Filamin A could play as a scaffold protein that mediates Rac1 activation. (iii) Activated Rac1 by Trio might enhance gastric cancer metastasis.

## Data Availability

The data presented in this study are available on request from the corresponding author.

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
