# Peer review of "RhoGDI2-Mediated Rac1 Recruitment to Filamin A Enhances Rac1 Activity and Promotes Invasive Abilities of Gastric Cancer Cells"

_cancers, 2022, doi:10.3390/cancers14010255_

Round 1
Reviewer 1 Report
In the manuscript by Kim et al. (Cancers 1492233) entitled “RhoGDI2-mediated Rac1 recruitment to Filamin A enhances Rac1 activity and promotes invasive abilities of gastric cancer cells” authors analyze the mechanism used by RhoGDI2 to activate Rac1 in gastric cancer cells. They find that its interaction with Filamin A is involved and would facilitate the action of the Rac1 GEF, Trio. In general, this study is interesting and uncovers a new mechanism of Rac1 activation that could be involved in gastric cancer metastasis. However, there are several weak points that need to be solved:
Major points:
1-While the interaction between endogenous Rac1 and Filamin A is shown to occur in MKN28 gastric cells, the interaction between RhoGDI2 and Rac1 is only demonstrated in HEK293 transfected with exogenous RhoGDI2 and Rac1. However, this last interaction should be also shown in gastric cancer cells with the endogenous proteins.
2-The authors do not explain why they chose only two gastric cell lines: SNU484 and MKN28, for this study. They use U484 cell line that does not express RhoGDI2, so they overexpress exogenous RhoGDI2. However, it is unclear the behavior of this cell line as compared with that of MKN28 cells in terms of invasion capacity, something that would be tested and/or explained considering that the authors conclude that the increase in Rac1 activity mediated by RhoGDI2/Filamin A enhances invasion. In addition, it would be important to use different gastric cancer cell lines with high and/or medium endogenous levels of RhoGDI2 (not only MKN28) and show that the mechanism of Rac1 activation described here through RhoGDI2/Filamin A/Trio/Rac1 also operates using a knock-down strategy.
3-In the Rac1 activity assays (fig. 2G, 3A, etc) it is only shown active Rac1 and total Rac1 in western-blots. It should be also shown the levels of actin or other normalizing protein and the quantification of active Rac1 versus total Rac1/actin levels.
4-The authors do not indicate in the figure legends the number of independent experiments performed. It is essential to include this information.
5-Figure 5 shows the analysis of RhoGDI2 and Filamin A levels present in samples from gastric cancer patients using public databases information comparing stage I-II to stages III-IV, but there is no comparison with samples from normal gastric tissue. This is important in order to know if there is also an increased expression in these tumors as compared with non-tumor gastric tissue.
Minor points:
1-The material and methods sections should be improved. It is necessary to include the following information:
-References of the antibodies used.
-In the migration and invasion assays, it is necessary to indicate if the cells placed in the upper chamber are in a medium with or without serum and the pore size of the membrane.
-Although there are previous published data about a potential interaction between RhoGDI2 and Filamin A, whose physiological relevance is unclear (Song et al. Biochem Biophys Res Commun. 2016;469(3):659-64), the authors don´t mention this work. This should be explained, either in the introduction or in the discussion.
Author Response
Response to Reviewer 1 Comments
Major points:
Point 1: While the interaction between endogenous Rac1 and Filamin A is shown to occur in MKN28 gastric cells, the interaction between RhoGDI2 and Rac1 is only demonstrated in HEK293 transfected with exogenous RhoGDI2 and Rac1. However, this last interaction should be also shown in gastric cancer cells with the endogenous proteins.
Response 1: We thank the reviewer for the constructive advice and completely agree with it as mentioned above. We confirmed the interaction between RhoGDI2 and Rac1 in RhoGDI2-overxpressing SNU484 cells and parental MKN28 cells. We added these data in Supplementary Figure 1.
Point 2: The authors do not explain why they chose only two gastric cell lines: SNU484 and MKN28, for this study. They use U484 cell line that does not express RhoGDI2, so they overexpress exogenous RhoGDI2. However, it is unclear the behavior of this cell line as compared with that of MKN28 cells in terms of invasion capacity, something that would be tested and/or explained considering that the authors conclude that the increase in Rac1 activity mediated by RhoGDI2/Filamin A enhances invasion. In addition, it would be important to use different gastric cancer cell lines with high and/or medium endogenous levels of RhoGDI2 (not only MKN28) and show that the mechanism of Rac1 activation described here through RhoGDI2/Filamin A/Trio/Rac1 also operates using a knock-down strategy.
Response 2: We described why we chose SNU484 and MKN28 cells in the ‘Results’ section (from lane 230 to 240). We completely agree with it as mentioned above. As reviewer said, a limitation of our study is the lack of experiments exploring the mechanism of Rac1 activation via RhoGDI2/Filamin A/Trio/Rac1 described in our study in various gastric cancer cell lines. However, please understand that there are limitations to gastric cancer cell lines that can be used to explore the Rac1 activation mechanism by RhoGDI2. In our previously published paper, we checked the expression level of RhoGDI2 in various gastric cancer cell lines, and found that the expression of RhoGDI2 was only observed in MKN28 and SNU638 cells, which have been known to be highly invasive gastric cancer cell lines, and was not expressed at all in other cell lines (Ref. 17). For this reason, the only cell lines we can use are MKN28 and SNU638 cells. However, in the case of SNU638 cells, for unknown reasons, stable clones in which RhoGDI2 expression was suppressed could not be created, so it could not be used in this study.
Ref. 17; Cho et al, RhoGDI2 expression is associated with tumor growth and malignant progression of gastric cancer. Clin. Cancer Res. 2009, 15, 2612-2619.
Point 3: In the Rac1 activity assays (fig. 2G, 3A, etc) it is only shown active Rac1 and total Rac1 in western-blots. It should be also shown the levels of actin or other normalizing protein and the quantification of active Rac1 versus total Rac1/actin levels.
Response 3: We have already shown the levels of α-tubulin in Figures (2G, 3A, and 4D) that have been assayed for Rac1 activity. We rearranged the blots in Figure 2G, 3A, and 4D, and quantified the Rac1 activity using α-tubulin and total Rac1 expression levels as controls.
Point 4: The authors do not indicate in the figure legends the number of independent experiments performed. It is essential to include this information.
Response 4: We completely agree with it as mentioned above. We described how many independent experiments we performed in the figure legends.
Point 5: Figure 5 shows the analysis of RhoGDI2 and Filamin A levels present in samples from gastric cancer patients using public databases information comparing stage I-II to stages III-IV, but there is no comparison with samples from normal gastric tissue. This is important in order to know if there is also an increased expression in these tumors as compared with non-tumor gastric tissue.
Response 5: We completely agree with it as mentioned above. To determine the differences in RhoGDI2 and Filamin A expression levels between normal and gastric cancer patients tissues, we analyzed additional public microarray data set (GSE54129). From these analyses, we found that expression of RhoGDI2 and FLNA was increased in gastric cancer tissues compared to normal tissues. We added these data in Figure 5A and 5B, respectively.
Minor points:
The material and methods sections should be improved. It is necessary to include the following information:
Point 1: References of the antibodies used.
Response 1: We added the catalog number of the antibodies we used in the ‘Materials and Methods’ section.
Point 2: In the migration and invasion assays, it is necessary to indicate if the cells placed in the upper chamber are in a medium with or without serum and the pore size of the membrane.
Response 2: For wound healing assay, we used normal medium (with 10% FBS). For invasion assay, we suspended cells in 250 ml of serum free media in the upper chamber containing 8 mm pore size membrane. We described about it in the ‘Materials and Methods’ section.
Point 3: Although there are previous published data about a potential interaction between RhoGDI2 and Filamin A, whose physiological relevance is unclear (Song et al. Biochem Biophys Res Commun. 2016;469(3):659-64), the authors don´t mention this work. This should be explained, either in the introduction or in the discussion.
Response 3: We discussed the paper published by Song et al, and were able to propose a good hypothesis based on this paper. We thank the reviewer for recommending the paper published by Song et al.
Reviewer 2 Report
Kim et al. revealed the mechanisms of gastric cancer metastasis through RhoGDI2-mediated Rac1 recruitment to Filamin A. The authors performed in vitro experiment and found the interaction of each molecule by using pulldown experiment and functional migration assay. However, I have some questionable points. Followings are my specific comments.
- Materials and methods should be described in more detail (e.g. contents of lysis buffer, the methods of construction of expressional vectors and transfection, the sequence of siTrio and transfection)
- Lane 200, the authors claim RhoGDI2⊿20 does not bind with FLNA. However, IP-Western blotting showed the binding of RhoGDI2⊿20 and Rac1 (Figure 1C).
- Figure 1C, Likewise, the blot of HA showed the contamination of each sample, because the blot of HA is detectable in the lysate of HA-RhoGDI2 non transduced HEK293 cells.
- Figure 2A, why GFP-Rac1 is undetectable in first lane? Is endogenous RhoGDI2 not working this experiment?
- Figure 4, how was the growth change in Trio knocked down cells?
- Figure 5, the expressional change of RhoGDI2 and FLNA in database analysis looks like small considering functional changes. It is difficult to certify the quality of data because the methods of sample quality control and data normalization are heterogeneity in database analysis. The authors should confirm the expressional abnormality using clinical sample of gastric cancer patients, at least cell lines, in author’s facility.
- The influence of RhoGDI2 in metastasis should be assessed in vivo animal model.
Author Response
Response to Reviewer 2 Comments
Point 1: Materials and methods should be described in more detail (e.g. contents of lysis buffer, the methods of construction of expressional vectors and transfection, the sequence of siTrio and transfection)
Response 1: We thank the reviewer for the constructive advice and completely agree with it as mentioned above. We described more detail for the contents of lysis buffer, gene cloning and transfection, and the siTrio sequences and transfection in the ‘Materials and Methods’ section.
Point 2: Lane 200, the authors claim RhoGDI2⊿20 does not bind with Rac1. However, IP-Western blotting showed the binding of RhoGDI2⊿20 and Rac1 (Figure 1C).
Response 2: We completely agree with it as mentioned above. We apologize for our misrepresentation. We changed our description ‘As shown in Figure 1, RhoGDI2△20 can bind to FLNA but not to Rac1’ to ‘As shown in Figure 1, when compared to Rac1 and WT-RhoGDI2, the binding between Rac1 and RhoGDI2△20 was markedly reduced’ in Lane 223.
Point 3: Figure 1C, Likewise, the blot of HA showed the contamination of each sample, because the blot of HA is detectable in the lysate of HA-RhoGDI2 non transduced HEK293 cells.
Response 3: We replaced the blots of HA in Figure 1C with the blots not contaminated.
Point 4: Figure 2A, why GFP-Rac1 is undetectable in first lane? Is endogenous RhoGDI2 not working this experiment?
Response 4: RhoGDI2 is not expressed in HEK293T cells (data not shown) as in the SNU484 gastric cancer cell line (Figure 2D), and then we could not detect any interaction between endogenous Filamin A and exogenously expressed GFP-Rac1 in HEK293T cells without expression of exogenous RhoGDI2.
Point 5: Figure 4, how was the growth change in Trio knocked down cells?
Response 5: We performed the proliferation assay with Trio-depleted cells and control cells, and found that all cells exhibited similar growth rates under the same growth conditions. We added these data in Figure 4G.
Point 6: Figure 5, the expressional change of RhoGDI2 and FLNA in database analysis looks like small considering functional changes. It is difficult to certify the quality of data because the methods of sample quality control and data normalization are heterogeneity in database analysis. The authors should confirm the expressional abnormality using clinical sample of gastric cancer patients, at least cell lines, in author’s facility.
Response 6: We completely agree with it as mentioned above. The clinical significance of RhoGDI2 expression in gastric cancer patients has already been confirmed and published in our previous paper (Ref. 17). In our previous report, we clearly showed that RhoGDI2 is frequently stained in the tumor cells of advanced stage gastric cancer tissues. In contrast, negative or very weak RhoGDI2 staining was observed in early stage gastric cancer tissues and normal gastric tissues. It would be great if the same experiments could be performed using Filamin A, but there are currently no gastric patient tissues accessible, thus staining normal gastric tissues and gastric cancer patient tissues for Filamin A expression is not possible.
We also investigated the expression level of RhoGDI2 in various gastric cancer cell lines and published in our previous paper (Ref. 17). In our previous report, we clearly showed that RhoGDI2 is expressed only in the gastric cancer cell lines derived from the secondary tumor sites (ascites), SNU16 and SNU638, but not in the primary cancer cell lines, SNU1, SNU484, and SNU719. RhoGDI2 was also expressed in the highly invasive gastric cancer line, MKN28, but no in MKN45 cells, which evidence poor invasive ability. We also examined the expression level of Filamin A in the same gastric cancer cell lines, but found that almost the same amount of Filamin A was expressed in all cell lines we tested (data not shown).
Ref. 17; Cho et al, RhoGDI2 expression is associated with tumor growth and malignant progression of gastric cancer. Clin. Cancer Res. 2009, 15, 2612-2619.
To have more confidence in our results, we investigated the expression levels of RhoGDI2 and Filamin A in normal gastric tissues and gastric cancer tissues through additional public microarray data set analysis (GSE54129). From these analyses, we found that expression of RhoGDI2 and FLNA was increased in gastric cancer tissues compared to normal tissues. We added these data in Figure 5A and 5B, respectively.
Point 7: The influence of RhoGDI2 in metastasis should be assessed in vivo animal model.
Response 7: The influence of RhoGDI2 in metastasis of gastric cancer cells in an in vivo animal model has already been performed and published in our previous paper (Ref. 17). In our previous paper, we clearly showed that RhoGDI2 expression profoundly increases gastric tumor metastasis in vivo.
Ref. 17; Cho et al, RhoGDI2 expression is associated with tumor growth and malignant progression of gastric cancer. Clin. Cancer Res. 2009, 15, 2612-2619.
Round 2
Reviewer 1 Report
In the new version of the manuscript by Kim et al. (Cancers 1492233) the authors have included a few new data and explanations according to the comments to the previous version. Therefore, the paper has been improved.
Author Response
Our manuscript has been carefully reviewed by an experienced editor whose first language is English and who specializes in editing papers written by scientists whose native language is not English.
Reviewer 2 Report
The authors adequately, if not completely, addressed my concerns. The authors must check followings before acceptance.
- Figure 3D and 4G legends, the numbers of plated cells are 1×10^5 cells?
- The authors must perform ANOVA analysis and post hoc test to assess the significance of 3 groups.
Author Response
Response to Reviewer 2 Comments
Minor points:
Point 1: Figure 3D and 4G legends, the numbers of plated cells are 1×10^5 cells?
Response 1: We appreciated the reviewer’s precise point. We changed the values of the y-axis in Figure 3D and 4G.
Point 2: The authors must perform ANOVA analysis and post hoc test to assess the significance of 3 groups.
Response 2: We have already performed ANOVA analysis and Dunnett’s post hoc tests to assess the significance of the 3 groups. We described this in the ‘Statistical Analysis’ section of ‘Materials and Methods’.